# Menopausal Hormone Therapy and Risk of Endometrial Cancer: A Systematic Review

**DOI:** 10.3390/cancers12082195

**Published:** 2020-08-06

**Authors:** Clemens B. Tempfer, Ziad Hilal, Peter Kern, Ingolf Juhasz-Boess, Günther A. Rezniczek

**Affiliations:** 1Department of Gynecology and Obstetrics, Ruhr-Universität Bochum, Bochum, Germany and Comprehensive Cancer Center of the Ruhr-Universität Bochum (RUCCC), 44708 Bochum, Germany; ziadhilal@hotmail.com (Z.H.); guenther.rezniczek@rub.de (G.A.R.); 2Department of Gynecology and Obstetrics, St. Elisabeth-Krankenhaus Bochum, 44787 Bochum, Germany; peter.kern@uk-essen.de; 3Department of Gynecology and Obstetrics, University of Freiburg, 79110 Freiburg, Germany; ingolf.juhasz-boess@uniklinik-freiburg.de

**Keywords:** endometrial cancer, hormone therapy, hormone replacement, estradiol, progestins, progesterone, hormone-dependent cancer

## Abstract

Background: Menopausal hormone therapy (MHT) is an appropriate treatment for women with the climacteric syndrome. The estrogen component of MHT effectively alleviates climacteric symptoms but also stimulates the endometrium and thus may increase the risk of endometrial cancer (EC). Materials and Methods: We performed a systematic literature search of the databases PubMed and Cochrane Central Register of Controlled Trials to identify controlled and uncontrolled clinical trials reporting on the prevalence and/or incidence of EC among women using MHT. Results: 31 publications reporting on 21,306 women with EC diagnosed during or after MHT were identified. A significantly reduced risk of EC among continuous-combined (cc)MHT users with synthetic progestins (SPs) was demonstrated in 10/19 studies with odds ratios (ORs)/hazard ratios (HRs) between 0.24 and 0.71. Only one study documented an increased risk of EC among long-term users (≥10 years), not confirmed in three other sub-group analyses of women with ≥6, ≥5, and >10 years of ccMHT use. A significantly increased risk of EC among users of sequential-combined (sc)MHT with SPs was demonstrated in 6/12 studies with ORs/HRs between 1.38 and 4.35. Number of days of progestin per month was a significant modulator of EC risk. A decreased risk of EC was seen in obese women. Two studies documented an increased risk of EC among users of cc/scMHT with micronized progesterone. A significantly increased risk of EC among estrogen-only MHT users was demonstrated in 9/12 studies with ORs/HRs between 1.45 and 4.46. The adverse effect of estrogen-only MHT was greatest among obese women. Conclusion: ccMHT with SPs reduces the risk of EC, whereas estrogen-only MHT increases the risk. scMHT with SPs and cc/scMHT with micronized progesterone increase the risk of EC depending on type of progestin, progestin dosage, and duration of MHT use.

## 1. Introduction

Endometrial cancer (EC) is a malignancy derived from the epithelial lining of the uterine cavity. It is the most common female pelvic malignancy with an estimated life-time incidence of 4% [1]. Typically, older women are affected with a mean age at the time of diagnosis of 69 years [2]. EC generally has a good prognosis and therefore accounts for only 2% of cancer-related deaths in females despite its high incidence and prevalence. The reason for the favorable outcome of most patients with EC is that the diagnosis of EC can be established in around 90% of cases at an early stage of disease [1,2]. The most characteristic pathophysiological feature of EC is its hormone-dependence. Endometrioid adenocarcinoma of the endometrium, often referred to as type-I EC, accounts for 85% to 90% of all ECs and is typically stimulated by the long-term exposure to unopposed estrogen [3]. In contrast, other subtypes of EC, e.g., serous carcinoma, serous-papillary carcinoma, clear cell carcinoma, and carcinosarcoma, are not hormone-dependent and have a poor prognosis. In the present review, we will focus on hormone-dependent, type-I EC and its association with exogenous stimulation by estrogens. Long-term, unbalanced exposure to estrogens may occur in two ways, either due to an internal progesterone deficit and a subsequent relative excess of estradiol, e.g., in women with the polycystic ovary syndrome (PCOS), or due to the intake of exogenous estrogens such as in women taking menopausal hormone therapy (MHT). The association between PCOS and EC will be investigated in detail in another review article in this Special Issue. Thus, in the present review, we will focus on the risk of EC among women taking MHT.

MHT is an appropriate therapy for peri- and postmenopausal women with signs and symptoms of the climacteric syndrome such as hot flushes, sweating, mood swings, vaginal dryness, dyspareunia, hair loss, eye dryness, i.e., sicca syndrome, and joint pain, i.e., arthropathia climacterica [3]. MHT typically contains estradiol as the estrogenic compound which effectively alleviates vasomotor symptoms such as hot flushes and night sweats and—to a lesser extent—other climacteric symptoms [3]. If, however, a woman treated with estradiol still has an intact uterus, the estrogenic compound of MHT will also stimulate endometrial growth irrespective of the woman’s age and may thus lead to the development of endometrial hyperplasia and ultimately EC. Therefore, it is the standard of care to combine the estrogenic compound used in MHT with a progestin [3].

Progestins used for MHT are either synthetic progestogenic compounds such as norethisterone acetate (NETA) and medroxyprogesterone acetate (MPA) or natural progesterone. Progestins antagonize the stimulating effect of estradiol on the endometrium via the progesterone receptors alpha (PR-A) and beta (PR-B). Specifically, the effect of progestins is generally considered to represent the combined activities of PR-A and PR-B. Upon ligand binding, PR-A and PR-B affect cellular function by altering gene expression via ligand-activated transcription factors or via Src tyrosine kinases in the cytoplasm to activate mitogen-activated protein kinases (MAPKs) which then affect estradiol receptor gene expression [4]. Progesterone receptor expression by uterine epithelial cells is stimulated by estrogens via the estrogen receptor-α (ER-α). Consequently, progesterone responsiveness of the endometrium is dependent on the presence of an estrogenic stimulus such as the one conferred by MHT [5].

The counterbalancing effect of progestin on endometrial stimulation is dependent on the type and dosage of the progestin used for MHT as well as the duration of MHT. For example, the continuous, daily use of a synthetic progestin in combination with estradiol over the course of 5 years has been shown to significantly reduce the risk of EC in the Women’s Health Initiative (WHI) trial [6]. Whereas the continuous daily use of a synthetic progestin decreases the risk of EC, the long-term use of a synthetic progestin in a sequential (non-daily) combination with estradiol has been shown to increase the risk of EC in some, but not in other studies [7]. In addition, the use of natural progesterone in the form of dydrogesterone or micronized progesterone in combination with estrogen over the course of 5 years has been shown to significantly increase the risk of EC in the E3N cohort study [8]. Together, these observations suggest that both the type of progestin used for MHT as well as the specific scheme of MHT have a significant impact on the risk of EC.

The aim of the present systematic review was to summarize the current knowledge on the risk of developing EC among women using different forms of MHT. In order to achieve this, we performed a systematic search of the literature using the PubMed database and the Cochrane Central Register of Controlled Trials to identify controlled and uncontrolled clinical trials reporting on the prevalence and/or incidence of EC among women using MHT.

## 2. Materials and Methods

We performed a systematic literature search of the databases PubMed and Cochrane Central Register of Controlled Trials using the following search terms: “hormone replacement therapy” MeSH Terms OR “hormone” All Fields AND “replacement” All Fields AND “therapy” All Fields OR “hormone replacement therapy” All Fields AND “endometrial neoplasms” MeSH Terms OR “endometrial” All Fields AND “neoplasms” All Fields OR “endometrial neoplasms” All Fields OR “endometrial” All Fields AND “cancer” All Fields OR “endometrial cancer” All Fields (search date: 2020-06-20). The methodology followed the PRISMA criteria. The PICO question was as follows: Does a hormone therapy with estrogens and/or estrogens combined with progestins increase the risk of endometrial cancer in peri- or postmenopausal women? Screening, eligibility, and data analysis were performed by two authors independently (CBT and GAR). Discrepancies were solved by consensus. Study investigators were not contacted to obtain further information. The main outcome of interest was the risk of EC in relation to different forms of peri- and postmenopausal MHT. Bias of individual studies and grading of the strength of the evidence were not assessed. With the above described search strategy, 1359 citations were identified. Therefore, the search was restricted to the last 20 years, i.e., January 2000 until May 2020 yielding 883 citations. After screening all abstracts, 29 appropriate citations were selected reporting on the incidence and prevalence of EC among women undergoing MHT. MHT was defined for the purpose of this systematic review as systemic (oral, transdermal or vaginal) use of estradiol or conjugated equine estrogens or estradiol or conjugated equine estrogens combined with a synthetic progestin (norethisterone acetate, medroxyprogesterone acetate, megestrol acetate, chlomadinone acetate, medrogestone, levonorgestrel, cyproterone acetate, drospirenone, dienogest) or combined with progesterone (natural progesterone or dydrogesterone) in a continuous combined therapy scheme (ccMHT) or in a sequentially-combined therapy scheme (scMHT). Studies not reporting individual patient data and studies containing no extractable clinical data were excluded. We also excluded studies reporting on MHT with dehydroepiandrosterone, tibolone, or androgens with or without estrogens (*n* = 854). All 29 appropriate citations were then retrieved in full and cross reference searching was performed identifying two further citations for a total of 31 appropriate citations reporting on the incidence and prevalence of EC among women using MHT published between January 2000 and May 2020. Figure 1 shows a flow diagram of the literature search algorithm. Data were extracted and analyzed in a descriptive manner. Meta-analysis was not performed due to the heterogeneity of studies.

## 3. Results

In a systematic literature search using the search criteria as described above (search date 20 June 2020), we identified 883 citations published after 1 January 2000. A total of 854 citations were excluded because they did not report on the incidence or prevalence of EC among women using MHT as defined for this review. Using the remaining 29 citations, cross reference searching identified two further appropriate citations. Thus, in summary, 31 citations reporting on the incidence and prevalence of EC among women using MHT were included in this review [6,7,8,9,10,11,12,13,14,15,16,17,18,19,20,21,22,23,24,25,26,27,28,29,30,31,32,33,34,35,36]. Among them, we found 15 cohort studies [8,9,10,11,12,14,16,17,21,23,24,25,27,30,32], 10 case–control studies [7,18,19,20,26,29,33,34,35,36], 2 randomized controlled trials [6,28], 2 narrative reviews [15,31], 1 meta-analysis [22], and 1 systematic review [13], describing a total of 21,306 patients with EC.

The clinical characteristics of the subset of 20 studies reporting individual patient data on EC among women using various forms of MHT are shown in Table 1, Table 2, Table 3 and Table 4. Specifically, the clinical characteristics of individual studies reporting on ccMHT with synthetic progestins and EC risk are shown in Table 1. The clinical characteristics of individual studies reporting on scMHT with synthetic progestins and EC risk are shown in Table 2. Table 3 shows the clinical characteristics of individual studies reporting on ccMHT and scMHT with progesterone or dydrogesterone and EC risk, and Table 4 shows the clinical characteristics of individual studies reporting on MHT with estrogens in women with an intact uterus and EC risk.

### 3.1. ccMHT with Synthetic Progestins Reduces The Risk of EC

The clinical characteristics of individual studies reporting on ccMHT with synthetic progestins and EC risk are shown in Table 1. In 8 case–control studies (CCS), 9 cohort studies (CS), and 2 randomized controlled trials (RCTs), 10,265 women with EC and 1,969,172 controls were analyzed. A significantly reduced risk of EC among ccMHT users with synthetic progestins was demonstrated in 9/19 studies with ORs/HRs between 0.24 [21] and 0.71 [30]. In the remaining studies, neither a risk reduction nor a risk increase was observed. Only one study documented an increased risk of EC in the sub-group of long-term users (≥10 years) of ccMHT with synthetic progestins [7], which was not confirmed in three other sub-group analyses of long-term users with ≥6 years of use [26], ≥5 years of use [27], and >10 years of use [19].

### 3.2. scMHT with Synthetic Progestins May Increase The Risk of EC

The clinical characteristics of individual studies reporting on scMHT with synthetic progestins and EC risk are shown in Table 2. In 6 CCS and 6 CS, 9663 women with EC and 1,843,377 controls were analyzed. A significantly increased risk of EC among scMHT users with synthetic progestins was demonstrated in 6/12 studies with ORs/HRs between 1.38 [19] and 4.35 [7]. In the remaining six studies, no effect on EC risk was observed. Only one study documented a decreased risk of EC in the sub-group of short-term users (<5 years) of ccMHT with synthetic progestins [19]. A beneficial effect of scMHT with synthetic progestins was also seen in the subgroup of obese women [30]. Number of days per month of progestin use was also a modulator of EC risk increase with a higher risk in case of <10 days/month [7,26,33].

### 3.3. ccMHT and scMHT with Progesterone or Dydrogesterone Increases The Risk of EC

The clinical characteristics of individual studies reporting on ccMHT and scMHT with progesterone or dydrogesterone and EC risk are shown in Table 3. Only two CS were identified [8,21] with 902 cases and 180,202 controls. Both studies documented an increased risk of EC among users of micronized progesterone (MP). Short-term use (≤5 years) of MP and ever use of dydrogesterone and progesterone derivatives were not associated with an increased risk of EC.

### 3.4. Estrogen-only MHT in Women with an Intact Uterus Increases the Risk of EC

The clinical characteristics of individual studies reporting on estrogen-only MHT in women with an intact uterus and EC risk are shown in Table 4. In 4 CCS and 8 CS, 10,241 women with EC and 1,952,004 controls were analyzed. A significantly increased risk of EC among estrogen-only MHT users was demonstrated in 9/12 studies with ORs/HRs between 1.45 [30] and 4.46 [7]. In the remaining three studies, no effect on EC risk was observed. No study documented a decreased risk of EC. The adverse effect of estrogen-only MHT was greatest in the subgroup of obese women [30]. Both past and current use and type of estradiol (estradiol, conjugated estrogens, non-conjugated estrogens) increased EC risk.

### 3.5. Narrative Reviews on MHT and EC Risk

In a narrative review of the literature, Brinton and Felix stated that in most studies including women who had ever used ccMHT (>25 days/months) EC risk was reduced relative to non-users (meta-analysis relative risk (RR) 0.78; 95% confidence interval (CI) 0.72–0.86 based on observational studies) [15]. The reduced risk was greatest among obese women. In contrast, women who had ever used scMHT with <10 days of progestins per month were at a significantly increased risk of EC with meta-analysis results showing an overall RR of 1.76 (95% CI 1.51–2.05). However, when progestins were given for 10–24 days/month, scMHT appeared unrelated to the risk of EC with an RR of 1.07 (95% CI 0.92–1.24). Sjögren et al. performed a systematic review of the literature and summarized 28 studies reporting on menopausal women with intact uteri treated with estrogen only or with estrogen plus progestin for a minimum of one year. [13]. They found that observational studies confirmed an increased risk among users of estrogen alone. Continuous combined therapy showed a lower risk than sequential combined therapy. The newer marketed micronized progesterone increased the risk notably, also when administered continuously.

A stepwise model of MHT and endometrial carcinogenesis with abnormal endometrial proliferation, atypical hyperplasia, and endometrial cancer was summarized by Horn et al. [31]. They report that about 15% of endometrial biopsies taken from women on scMHT show abnormal proliferative activity with atypical endometrial hyperplasia identified in 1% of cases. In contrast, endometrial biopsies from women using ccMHT typically demonstrate endometrial atrophy with only 2–3% of biopsies showing abnormal proliferative activity but without atypical hyperplasia. The risk of atypical hyperplasia or carcinoma under unopposed estrogen-mono MHT varies from 2 to 10%.

### 3.6. Body Mass Index, MHT, and EC Risk

In a meta-analysis of nine studies analyzing EC risk by body mass index (BMI) and MHT use, Crosbie et al. found that ever use of MHT reduced the risk of EC conferred by being overweight [22]. Specifically, the overall risk ratio of EC per 5 kg/m^2^ increase in BMI was 1.60 (95% CI 1.52–1.68). Among never users of MHT, the overall risk ratio of EC per 5 kg/m^2^ increase in BMI was 1.90 (95% CI 1.57–2.31) compared to only 1.18 (95% CI 1.06–1.31) among ever users of MHT. This analysis did not differentiate between ccMHT and scMHT. Similarly, in a cohort study of 103,882 postmenopausal women with 677 cases of EC, Chang et al. also found that MHT significantly modified the relations of BMI and EC risk [25]. The RR of EC for a base-line BMI ≥30 versus < 25 kg/m^2^ was 5.41 (95% CI 4.01–7.29) among non-users of MHT and was markedly reduced to 2.53 (95% CI 1.21–5.30) among former users of MHT. In line with these findings, another large cohort study with 33,436 postmenopausal women and 318 EC cases confirmed that BMI was a strong predictor of EC risk (RR 4.41; 95% CI 2.7–7.2 for BMI ≥ 35 kg/m^2^), but this association was no longer evident among ever users of cc/scMHT [24].

### 3.7. Influence of MHT on EC-Specific Mortality

In a sub-analysis of the National Institutes of Health-American Association of Retired Persons (NIH-AARP) Diet and Health Study, the effect of MHT on the 10-year all-cause EC-specific mortality was assessed based on 890 EC cases [14]. Interestingly, pre-diagnosis use of MHT had a significant and variable effect on mortality depending on MHT type. For example, pre-diagnosis use of ccMHT or scMHT was associated with a significantly lower 10-year all-cause mortality (HR 0.65, 95% CI 0.43–0.99) and EC-specific mortality (HR 0.51, 95% CI 0.26–0.98) compared with never users. In contrast, former estrogen-only MHT users had a significantly higher all-cause mortality (HR 1.71, 95% CI 1.02–2.88) and EC-specific mortality (HR 2.17, 95% CI 0.96–4.90). A case–control study from Sweden also investigated the influence of MHT on EC-specific mortality and found significantly improved survival rates for ever users of any form of MHT (relative survival ratio [RER] 0.40; 95% CI = 0.16–0.97), in particular ever users of any form of estrogens (RER 0.38; 95% CI = 0.15–0.99) [23].

### 3.8. Genetic Modulation of the Association between MHT and EC

EC risk conferred by MHT may be modulated by the carriage of polymorphisms in genes encoding enzymes involved in sex steroid metabolism. McKean-Cowdin et al. examined the association between EC risk and estrogen-mono MHT by CYP17 genotype using 51 incident cases and 391 randomly selected controls from a multiethnic cohort in Hawaii and Los Angeles, California [34]. The risk of EC was significantly higher for women homozygous for the CYP17 T allele (OR 4.10; 95% CI 1.64–10.3), but not for women with the C allele (OR 1.31; 95% CI 0.53–3.21), suggesting that CYP17 variants affecting biosynthesis and metabolism of estrogen are potential individual modulators of EC susceptibility due to MHT.

An additional genetic variant modulating the risk of EC among MHT users has been described by Razawi et al. [18]. They found that a variation in the CYP11A1 gene may modify the risk of postmenopausal EC conferred by estrogen-only MHT. Specifically, in a nested case–control study within the California Teachers Study, 286 EC cases and 488 controls were genotyped for polymorphisms in six sex steroid metabolism genes. The strongest interaction was observed between duration of estrogen-only MHT and haplotype 1A of CYP11A1 (*p* = 0.010). The OR for EC per copy of haplotype 1A was 2.00 (95% CI: 1.05–3.96) for long-term estrogen-only MHT users. All other genetic variants were not associated with any form of MHT after correction for multiple testing.

### 3.9. Circulating Estrogen Levels during MHT and EC Risk

In a nested case–control study among postmenopausal women using MHT at baseline in the Women’s Health Initiative Observational Study (230 endometrial cancers, 253 controls), EC risk was unrelated to estrogen or estrogen metabolite levels among women who took ccMHT [10]. This is an interesting finding because it is counter-intuitive that circulating estrogens do not influence EC risk among women with MHT-induced high-estrogen levels.

### 3.10. MHT and Epidemiology of EC

In an interesting analysis of the Surveillance, Epidemiology, and End Results (SEER) data base, Constantine et al. noted an increase in EC incidence after the publication of the WHI Study coinciding with a decrease in ccMHT prescriptions in the US [9]. In detail, EC rates were constant from 1992 to 2002 but increased by 2.5% annually with a 10% increase from 2006 to 2012. Use of approved prescription ccMHT products decreased after the publication of the WHI data, whereas other risk factors either remained constant or decreased during the same time. Based on these observations, the authors concluded that the EC rate increase after the first publication of WHI data in 2002 may be associated with the decreased use of approved ccMHT. In accordance with these data, Wartko et al. confirmed that EC incidence rates increased in the US after 2002 [17]. In contrast to the constant EC rate pattern observed from 1992 to 2002 (annual percentage change: 0.0%), rates increased after 2002 in women 50–74 years old (annual percentage change: 2.5%) and in women 20–49 years old (annual percentage change: 2.1%). Post-2002 increases in incidence among women ages 50–74 were specific to Type I endometrial tumors.

## 4. Discussion

MHT is an established means to treat signs and symptoms of the climacteric syndrome but may affect the risk of developing EC. In a systematic review of the literature we identified 31 controlled and uncontrolled clinical studies assessing the risk of EC among women with different forms of MHT. In summary, the available evidence shows that MHT influences the risk of developing EC in many ways. Specifically, MHT with estrogens only, either estradiol or conjugated equine estrogens, increases the risk of EC in women with an intact uterus [7,8,11,12,16,21,26,27,29,30,32,35]. In contrast, ccMHT with synthetic progestins reduces the risk of EC [6,7,11,12,16,19,20,21,24,25,26,27,28,29,30,32,33,35,36]. scMHT with synthetic progestins [7,12,16,19,21,25,26,27,29,30,33,35] and ccMHT and scMHT with natural progestins increases the risk of EC depending on progestin dosage and duration of use [8,21].

ccMHT or scMHT is the standard of care for women with an intact uterus in order to avoid endometrial stimulation with subsequent endometrial hyperplasia and EC [4]. Our systematic review confirms that women with an intact uterus should not be treated with estrogen-only MHT. Independent of the woman’s age, the endometrial epithelium will start to proliferate when estrogens without adequate progestin-opposition are being used and the risk of developing EC will be significantly increased as demonstrated in >10,000 women with EC. The risk of EC while using estrogen-only MHT is greatest in obese women, who therefore represent an especially vulnerable population. Our review also confirms that ccMHT with synthetic progestins is safe regarding endometrial proliferation and carcinogenesis. Substitution of synthetic progestins with natural progesterone such as MP, however, is problematic based on the data in the literature. Our analysis documents an increased risk of EC among users of MP. If MP is being used as part of ccMHT or scMHT, short-term use for ≤5 years is recommended. In addition, care should be taken that ccMHT is used instead of scMHT with an adequate dosage of at least 200 mg/day based on the results of the Postmenopausal Estrogen/Progestin Interventions (PEPI) trial [37].

The most controversial issue identified in our review is the potentially elevated risk of EC in women using scMHT. Our analysis based on >9000 EC cases shows that half of all studies found a significantly elevated risk of EC with ORs/HRs between 1.3 and 4.3. Number of days per month of progestin use was an important modulator of EC risk increase with a higher risk in case of <10 days/month. Therefore, women with climacteric symptoms seeking medical care should be counselled that scMHT is probably best avoided completely. There is no physiological or medical advantage of scMHT compared to ccMHT [3]. Thus, ccMHT, which clearly does not increase the risk of EC, is a reasonable alternative to scMHT avoiding the uncertainty of a potentially increased risk of EC.

The strongest effect size regarding an increase in EC risk among users of MHT with estrogens only was documented by Doherty et al. with an OR of 11 for long-term users with ≥6 years of therapy [26]. The strongest effect size regarding an increase in EC risk among scMHT users was observed by Razawi for long-term users with an OR of 4.35 [7], whereas the strongest effect regarding a decrease in EC risk among ccMHT users was found by Allen et al. with a HR of 0.24 among ever users [21].

Of note, epidemiological data strongly suggest that MHT has a primary preventive effect regarding EC. Two independent analyses of US population-based data found a significant increase in EC incidence coinciding with a decrease in MHT prescriptions. Specifically, Constantine et al. noted that EC rates were constant from 1992 to 2002 but increased by 2.5% annually with a 10% increase from 2006 to 2012 [9]. In accordance with these data, Wartko et al. also confirmed that EC incidence rates increased in the US by 2.5% annually after 2002 [17]. Since ccMHT is especially effective for EC risk reduction among obese women, ccMHt should be recognized as an efficient preventive measure in light of the ongoing obesity epidemic.

The present review adds to our knowledge of EC as an unwanted side effect of MHT providing an up-to-date analysis of the evidence. The difference of the present review compared to other studies is its comprehensive design including narrative reviews and work related to genetic analyses, interaction with BMI, and the effects of MHT prescription practice on EC incidence. In addition, the latest review on this topic dates back to 2016 and thus, an update of the evidence related to this clinically relevant topic is appropriate.

The available data do not show that MHT-associated ECs have a specific spectrum of histological subtypes. In this regard, most MHT-associated ECs are thus endometrioid adenocarcinomas [7,8,19,21,25,30]. A preference for aggressive histological subtypes such as clear cell carcinomas or malignant mixed müllerian tumors, which has been noted for tamoxifen-associated ECs [1,2], has not been found when women using MHT are diagnosed with EC.

Limitations of this review include the lack of meta-analysis, heterogeneity of data, and restriction to the last 20 years omitting older studies. However, based on the broad spectrum of studies identified, we believe that this review is a comprehensive and representative assessment of the current state of knowledge regarding this topic. In the future, the association between genetic markers such as variants of CYP17 and CYP11A1 and EC risk among MHT users should be tested in further studies. Individualization of risk profiles by genetic assessment carries a high potential for improvement and personalization of medical care. However, at this time, genetic testing remains experimental, as demonstrated by the available literature identified in this review. Thus, there is no reason to recommend any form of genetic testing in MHT users except for thrombophilia testing (prothrombin G20210A mutation or factor V Leiden mutation) in women with a personal and/or family history of thrombosis [3].

## 5. Conclusions

In conclusion, MHT affects the risk of EC in different ways. MHT with estrogens only increases the risk of EC. We also found that type and duration of progestin use in the context of MHT is an important factor influencing a woman’s risk of developing EC. Specifically, ccMHT with synthetic progestins reduces the risk of EC, whereas scMHT with synthetic progestins and ccMHT and scMHT with natural progestins increases the risk of EC depending on progestin dosage and duration of use.

## Figures and Tables

**Figure 1 cancers-12-02195-f001:**
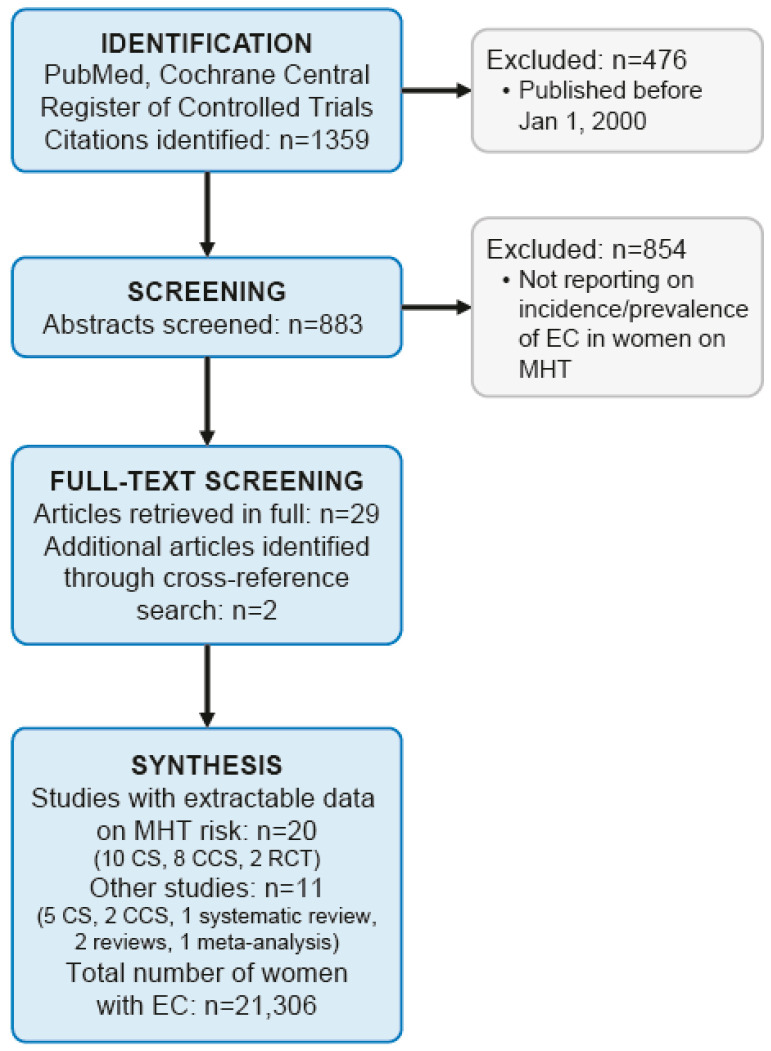
Flow diagram of the literature search algorithm.

**Table 1 cancers-12-02195-t001:** Clinical Characteristics of Studies Reporting on Continuous Combined Menopausal Hormone Therapy (ccMHT) with Synthetic Progestins and Endometrial Cancer (EC) Risk.

Author	Year	Study Type	Number of EC Patients	Number of Controls or Cohort Size	Population Characteristics	Tumor Characteristics	Risk of EC	Subgroup Analyses
Sponholtz [11]	2018	CS	300	47,555	US residents; Participants of the Black Women’s Health Study	Incident EC cases identified via self-report and questionnaires; confirmation by case records or cancer registries	IRR 1.55; 95% CI 0.76–3.11 for current ccMHT users vs. never users	IRR 0.63; 95% CI 0.36–1.09 for past ccMHT users vs. never users
Chlebowski [6]	2016	RCT	161 (ccMHT: 66; placebo: 95)	16,608 (8506 randomized to ccMHT; and 8102 to placebo)	Participants of the WHI study; postmenopausal US women with an intact uterus enrolled at 40 US clinical centers; median MHT duration: 5.6 years; median follow-up: 13 years	All histological types of EC included; all EC cases were centrally reviewed	HR 0.65; 95% CI 0.48–0.89	HR 0.77; 95% CI 0.45–1.31 during MHT HR 0.59; 95% CI 0.40–0.88 post MHT HR 0.42; 95% CI 0.15–1.22 for mortality due to EC
Mørch [12]	2016	CS	6202	914,595	All Danish women aged 50–79 years without previous cancer or hysterectomy from 1995–2009; data acquisition by National Prescription Register and National Cancer Registry	Incident EC cases including type I and type II ECs	RR 1.02; 95% CI 0.87–1.20 for ever users of ccMHT vs. never users	RR 0.45; 95% CI 0.20–1.01 for type II ECs for ever users of ccMHT vs. never users
Trabert [16]; Update of [25,27]	2013	CS	885	68,419	Members of American Association of Retired Persons (AARP) from 6 US states; data acquisition by repeated questionnaires	Incident EC cases identified from state-specific cancer registries and National Death Index	RR 0.64; 95% CI 0.49–0.83 for ever use of ccMHT vs. never use	RR 0.38; 95% CI 0.16–0.91 for former use RR 0.69; 95% CI 0.52–0.90 for current use
Jaakola [19]	2011	CCS	7261 (127)	19,490 (585)	Finnish Women with EC; data acquisition via Finnish Cancer Registry; Controls 3:1 per case from the Finnish National Population Register; use of MHT assessed via Finnish Medical Reimbursement Register	All histological types of EC included	OR 0.45; 95% CI 0.27–0.73 for use <5 years of ccMHT	OR 0.57; 95% CI 0.37–0.88 for 5–10 years OR 0.79; 95% CI 0.61–1.02 for 10+ years
Phipps [20] Update of [26,33,36]	2011	CCS	864(90)	1343 (227)	US residents in 3 counties of Washington state; cases identified via registry (Cancer Surveillance System); controls recruited via random-digit-dialing telephone interview; population restricted to ccMHT users	All EC cases were included except for in situ and non-epithelial cases	OR 0.50; 95% CI 0.37–0.67 for users of ccMHT vs. never users	OR 0.37; 95% CI 0.21–0.66 for long-term users (≥10 years) of ccMHT vs. never users. Protective effect of ccMHT most pronounced among obese women (BMI≥30): OR 0.19; 95% CI 0.05–0.68
Allen [21]	2010	CS	601	115,474	Participants of the European Prospective Investigation into Cancer and Nutrition (EPIC) study from 10 European countries	All incident EC cases were included based on self-report by questionnaire; follow-up data obtained via national cancer registries or insurance databases	HR 0.24; 95% CI 0.08–0.77 for ever use of ccMHT vs. never users	-
Razawi [7]	2010	CCS	311 (118)	570 (223)	Public school teachers and administrators; residence in California; data acquisition via California Cancer Registry	All histological types of EC included; exclusion of: in-situ carcinomas, endometrial sarcomas and mixed Müllerian tumors	OR 0.86; 95% CI 0.55–1.35 for <5 years OR 0.81; 95% CI 0.48–1.37 for 5–9 years	OR 2.05; 95% CI 1.27–3.30 for long-term (≥10 years) of ccMHT with ≥25 d/m progestin
McCullough [24]	2008	CS	318	33,436	Participants of the Cancer Prevention Study II Nutrition Cohort from 21 US states; data acquisition by repeated questionnaires and state-specific cancer registries	Incident EC cases identified via self-report and confirmed by hospital records or state-specific cancer registries and national Death Index	RR 4.41; 95% CI 2.70–7.20 for BMI ≥35 no longer significant among cc/scMHT users	Greater BMI (≥30) increased both risk of type I-EC (RR 4.22; 95% CI 3.07–5.81) and type II-EC (RR 2.87; 95% CI 1.59–5.16)
Chang [25]	2007	CS	677	103,882	Members of American Association of Retired Persons (AARP) from 6 US states; data acquisition by repeated questionnaires	Incident EC cases identified from state-specific cancer registries and National Death Index	RR 1.37; 95% CI 0.66–2.82 for obese ccMHT users vs. normal weight non MHT users	-
Doherty [26] Update of [33,36]	2007	CCS	1038 (52)	1453 (138)	US residents in 3 counties of Washington state; cases identified via registry (Cancer Surveillance System); controls recruited via random-digit-dialing telephone interview	All EC cases were included except for in situ and non-epithelial cases	OR 0.59; 95% CI 0.40–0.88 for ever users of ccMHT vs. never users	OR 0.77; 95% CI 0.45–1.3 for long-term use (≥6 years) of ccMHT vs. never users
Lacey [27]	2007	CS	433	73,211	Members of American Association of Retired Persons (AARP) from 6 US states; data acquisition by repeated questionnaires	Incident EC cases identified from state-specific cancer registries and National Death Index	RR 0.80; 95% CI 0.55–1.15 for ever users of ccMHT (≥20 days of progestin per cycle) vs. never users	Long duration of ccMHT (≥5 years) had also no effect (RR 0.85; 95% CI 0.53–1.36)
Strom [29]	2006	CCS	511 (53)	1412 (236)	US residents in Philadelphia region from 61 clinical centers; controls recruited via random-digit-dialing telephone interview	Only adenocarcinomas of the endometrium included; excluded: mixed Müllerian tumors, sarcomas, undifferentiated carcinomas, and squamous cell carcinomas	OR 0.69; 95% CI 0.48–0.99 for ever use of ccMHT	OR 0.77; 95% CI 0.33–1.81 for ever use of ccMHT vs. ever use of scMHT
Beral [30]	2005	CS	1320	716,738	UK residents participating in the Million Women Study; data acquisition by repeated questionnaires	Incident EC cases were prospectively identified; median follow-up was 3.4 years	RR 0.71; 95% CI 0.56–0.90 for ever users of ccMHT vs. never users	Beneficial effect of ccMHT was greatest among obese women
Bakken [32]	2004	CS	75	27,621	National, population-based cohort from Norway; postmenopausal women	Incident EC cases identified from questionnaires and followed-up using a national cancer registry; no restrictions on EC diagnosis	RR 0.7; 95% CI 0.4–1.4 for ever users of ccMHT vs. never users	-
Reed [33] Update of [36]	2004	CCS	647 (38)	1209 (123)	US residents in 3 counties of Washington state; cases identified via registry (Cancer Surveillance System); controls recruited via random-digit-dialing telephone interview	All EC cases were included except for in situ and non-epithelial cases	OR 0.5; 95% CI 0.3–0.7 for ever users of ccMHT (≤75 mg MPA/m) vs. never users (28 cases vs. 101 controls)	OR 0.9; 95% CI 0.4–1.9 for ever users of ccMHT (>75 mg MPA/m) vs. never users (10 cases vs. 22 controls)
Hill [36]	2000	CCS	969 (9)	1325 (33)	US residents in 3 counties of Washington state; cases identified via registry (Cancer Surveillance System); controls recruited via random-digit-dialing telephone interview	All EC cases were included except for in situ and non-epithelial cases	OR 0.6; 95% CI 0.3–1.3 for ever users of ccMHT vs. never users	OR 0.4; 95% CI 0.2–1.1 for ever use of ccMHT vs. ever use of scMHT
Samsioe [28]	2006	RCT	0	406 (3:1 transdermal vs. oral ccMHT)	Multicenter RCT in Sweden, Austria, Switzerland	Incident EC cases identified during 48-week follow-up	RR 0.0; no cases of endometrial hyperplasia or EC observed	-
Jain [35]	2000	CCS	512 (15)	513 (14)	Ontario, Canada	EC diagnosis restricted to adenocarcinoma, carcinoma, cystadenocarcinoma, or mixed Müllerian carcinoma; cases identified by Ontario Cancer Registry	OR 1.51; 95% CI 0.67–3.42 for ever use of ccMHT vs. never use	-
*Pooled Analysis*	-	CCS (*n* = 8), CS (*n* = 9), RCT (*n* = 2)	11,474 (10,265 after exclusion of update studies)	2,119,524 (1,969,172 after exclusion of update studies)	-	-	No risk increase: 8 studies Reduced risk: 9 studies	No risk increase in long-term users: 3 studies Increased risk in long-term users: 1 study

Abbreviations: ccMHT, continuous-combined menopausal hormone therapy; EC, endometrial cancer; CCS, case–control study; OR, odds ratio; CI, confidence interval; RCT, randomized controlled trial; WHI, Women’s Health Initiative; HR, hazard ratio; scMHT, sequential-combined menopausal hormone therapy; CS, cohort study; BMI, body mass index; UK, United Kingdom; RR, relative risk; MPA, medroxyprogesterone acetate; IRR, incidence rate ratio.

**Table 2 cancers-12-02195-t002:** Clinical Characteristics of Studies Reporting on scMHT with Synthetic Progestins and EC Risk.

Author	Year	Study Type	Number of EC Patients	Number of Controls or Cohort Size	Population Characteristics	Tumor Characteristics	Risk of EC	Subgroup Analyses
Mørch [12]	2016	CS	6202	914,595	All Danish Women aged 50–79 years without previous cancer or hysterectomy from 1995–2009; data acquisition by National Prescription Register and National Cancer Registry	Incident EC cases including type I and type II ECs	RR 2.06; 95% CI 1.88–2.27 for ever users of scMHT vs. never users	RR 1.81; 95% CI 1.2–2.81 for type II ECs for ever users of scMHT vs. never users
Trabert [16]; Update of [25,27]	2013	CS	885	68,419	Members of American Association of Retired Persons (AARP) from 6 US states; data acquisition by repeated questionnaires	Incident EC cases identified from state-specific cancer registries and National Death Index	RR 1.23; 95% CI 0.96–1.57 for ever users of scMHT vs. never users	RR 0.90; 95% CI 0.64–1.26 for <10 years RR 1.88; 95% CI 1.36–2.60 for ≥10 years Increased risk seen only among normal-weight women (BMI < 25)
Jaakola [19]	2011	CCS	7261 (422)	19,490 (1126)	Finnish Women with EC; data acquisition via Finnish Cancer Registry; Controls 3:1 per case from the Finnish National Population Register; use of MHT assessed via Finnish Medical Reimbursement Register	All histological types of EC included	OR 1.38; 95% CI 1.15–1.66 for 10+ years	OR 0.67; 95% CI 0.52–0.86 for <5 years OR 1.11; 95% CI 0.87–1.41 for 5–10 years
Allen [21]	2010	CS	601	115,474	Participants of the European Prospective Investigation into Cancer and Nutrition (EPIC) study from 10 European countries	All incident EC cases were included based on self-report by questionnaire; follow-up data obtained via national cancer registries or insurance databases	HR 1.52; 95% CI 1.00–2.29 for ever users of scMHT vs. never users	-
Razawi [7]	2010	CCS	311(50)	570(80)	Public school teachers and administrators; residence in California; data acquisition via California Cancer Registry	All histological types of EC included; exclusion of: in-situ carcinomas, endometrial sarcomas and mixed muellerian tumors	OR 4.35; 95% CI 1.68–11.22 for long-term (≥10 years) of scMHT with <10 d/m progestin	OR 1.70; 95% CI 1.28–2.31 per 5 years of MHT with <10 d/m progestin OR 1.10; 95% CI 0.75–1.55 per 5 years of MHT with 10–24 d/m progestin
Chang [25]	2007	CS	677	103,882	Members of American Association of Retired Persons (AARP) from 6 US states; data acquisition by repeated questionnaires	Incident EC cases identified from state-specific cancer registries and National Death Index	RR 2.20; 95% CI 1.01–4.82 for obese scMHT users vs. normal weight non MHT users	-
Doherty [26] Update of [33]	2007	CCS	1038 (109)	1453 (166)	US residents in 3 counties of Washington state; cases identified via registry (Cancer Surveillance System); controls recruited via random-digit-dialing telephone interview	All EC cases were included except for in situ and non-epithelial cases	OR 2.0; 95% CI 1.2–3.5 for ever users of scMHT (10–24 d/m progestin) for ≥ 6 years vs. never users	No increased risk for scMHT (10–24 d/m progestin; <6 years) vs. never users Increased risk for any duration of scMHT with <10 d/m of progestin (OR 5.9; 95% CI 2.9–12 for ≥6 years)
Lacey [27]	2007	CS	433	73,211	Members of American Association of Retired Persons (AARP) from 6 US states; data acquisition by repeated questionnaires	Incident EC cases identified from state-specific cancer registries and National Death Index	RR 0.74; 95% CI 0.39–1.40 for ever users of scMHT users (10–14 days of progestin per cycle) vs. never users	Long duration of scMHT (≥5 years) had also no effect (RR 0.79; 95% CI 0.38–1.66)
Strom [29]	2006	CCS	511 (9)	1412 (29)	US residents in Philadelphia region from 61 clinical centers; controls recruited via random-digit-dialing telephone interview	Only adenocarcinomas of the endometrium included; excluded: mixed muellerian tumors, sarcomas, undifferentiated carcinomas, and squamous cell carcinomas	OR 0.89; 95% CI 0.39–2.05 for ever users of scMHT	-
Beral [30]	2005	CS	1320	716,738	UK residents participating in the Million Women Study; data acquisition by repeated questionnaires	Incident EC cases were prospectively identified; median follow-up was 3.4 years	RR 1.05; 95% CI 0.91–1.22 for ever users of scMHT vs. never users	Beneficial effect of scMHT was greatest among obese women
Reed [33]	2004	CCS	647 (71)	1209 (129)	US residents in 3 counties of Washington state; cases identified via registry (Cancer Surveillance System); controls recruited via random-digit-dialing telephone interview	All EC cases were included except for in situ and non-epithelial cases	OR 0.8; 95% CI 0.5–1.4 for ever users of scMHT (<100 mg MPA/m) vs. never users (19 cases vs. 53 controls)	OR 1.0; 95% CI 0.6–1.7 for ever users of scMHT (≥100 mg MPA/m) vs. never users (24 cases vs. 57 controls) OR 4.2; 95% CI 2.0–8.9 for ever users of scMHT (<70 mg MPA/m; <10 days progestin/m) vs. never users (22 cases vs. 12 controls)
Jain [35]	2000	CCS	512 (65)	513 (87)	Ontario, Canada	EC diagnosis restricted to adenocarcinoma, carcinoma, cystadenocarcinoma, or mixed Mullerian carcinoma; cases identified by Ontario Cancer Registry	OR 1.05; 95% CI 0.71–1.56 for ever users of scMHT vs. never users	OR 1.49; 95% CI 0.93–2.40 for ≥3 years
*Pooled Analysis*	-	CCS (*n* = 6), CS (*n* = 6)	10,844 (9663 after exclusion of update studies)	1,993,936 (1,843,377 after exclusion of update studies)	-	-	No risk increase: 6 studies Increased risk: 6 studies	Increased risk for long-term use and/or short duration of progestins/month: 8 studies

Abbreviations: scMHT, sequential-combined menopausal hormone therapy; EC, endometrial cancer; CCS, case–control study; OR, odds ratio; CI, confidence interval; RR, relative risk; UK, United Kingdom; US, United States; MPA, medroxyprogesterone acetate; HR, hazard ratio; BMI, body mass index.

**Table 3 cancers-12-02195-t003:** Clinical Characteristics of Studies Reporting on ccMHT or scMHT with Progesterone or Dydrogesterone and EC Risk.

Author	Year	Study Type	Number of EC Patients	Number of Controls	Patient Population	Tumor Characteristics	Risk of EC	Subgroup Analyses
Fournier [8]	2014	CS	301	65,630	Postmenopausal French women insured by a National Health Insurance Fund; mainly teachers and their family members; residence in continental France; mean number of days with progesterone/month: 22.5 MP; 23.5 (dydrogesterone)	All histological types of EC included based on self-report; incident EC cases confirmed in 91% by local pathology report	HR 1.80; 95% CI 1.38–2.34 for ever use of ccMHT with MP HR 1.05; 95% CI 0.76–1.45 for ever use of ccMHT with dydrogesterone	HR 1.39; 95% CI 0.99–1.97 for≤5 years of ccMHT with MP HR 2.66; 95% CI 1.87–3.77 for>5 years of ccMHT with MP HR 1.69; 95% CI 1.06–2.70 for>5 years of ccMHT with dydrogesterone
Allen [21]	2010	CS	601	115,474	Participants of the European Prospective Investigation into Cancer and Nutrition (EPIC) study from 10 European countries	All incident EC cases were included based on self-report by questionnaire; follow-up data obtained via national cancer registries or insurance databases	HR 2.42; 95% CI 1.53–3.83 for ever use of MP vs. never use (sc/ccMHT not specified)	HR 1.23; 95% CI 0.84–1.79 for ever use of progesterone derivatives vs. never use (sc/ccMHT not specified)
*Pooled Analysis*	-	CS (*n* = 2)	902	180,202	-	-	Increased risk for MP: 2 studies No increased risk for dydrogesterone: 1 study	No increased risk for short-term MHT with MP: 1 study Increased risk for long-term use (>5 years) of dydrogesterone: 1 study

Abbreviations: ccMHT, continuous-combined menopausal hormone therapy; scMHT, sequential-combined menopausal hormone therapy; CS, cohort study; EC, endometrial cancer; MP, micronized progesterone; HR, hazard ratio; CI, confidence interval.

**Table 4 cancers-12-02195-t004:** Clinical Characteristics of Studies Reporting on Estrogen-Only MHT in Women with an Intact Uterus and EC Risk.

Author	Year	Study Type	Number of EC Patients	Number of Controls	Patient Population	Tumor Characteristics	Risk of EC	Subgroup Analyses
Sponholtz [11]	2018	CS	300	47,555	US residents; Participants of the Black Women’s Health Study	Incident EC cases identified via self-report and questionnaires; confirmation by case records or cancer registries	IRR 3.78; 95% CI 1.69–8.43 for current estrogen-only MHT users vs. never users	IRR 0.87; 95% CI 0.44–1.72 for past estrogen-only MHT users vs. never users
Mørch [12]	2016	CS	6202	914,595	All Danish Women aged 50–79 years without previous cancer or hysterectomy from 1995–2009; data acquisition by National Prescription Register and National Cancer Registry	Incident EC cases including type I and type II ECs	RR 2.70; 95% CI 2.41–3.02 for ever users of estrogen-only MHT vs. never users	RR 1.43; 95% CI 0.85–2.41) for type II ECs for ever users of estrogen-only MHT vs. never users
Fournier [8]	2014	CS	301	65,630	Postmenopausal women insured by a National Health Insurance Fund; mainly teachers and their family members; residence in continental France; type of estrogen: estradiol (92.8%), CEE (2.2%)	All histological types of EC included based on self-report; incident EC cases confirmed in 91% by local pathology report	HR 1.80; 95% CI 1.31–2.49 for ever use of estrogen-only MHT	HR 1.81; 95% CI 1.27–2.58 for≤5 years of estrogen-only MHT HR 3.53; 95% CI 1.44–8.66 for>5 years of estrogen-only MHT
Trabert [16]; Update of [27]	2013	CS	885	68,419	Members of American Association of Retired Persons (AARP) from 6 US states; data acquisition by repeated questionnaires	Incident EC cases identified from state-specific cancer registries and National Death Index	RR 1.13; 95% CI 0.88–1.46 for ever use of estrogen-only MHT vs. never use	RR 0.74; 95% CI 0.52–1.04 for <5 years RR 1.44; 95% CI 0.68–3.03 for 5–9 years RR 3.93; 95% CI 2.62–5.89 for ≥10 years
Allen [21]	2010	CS	601	115,474	Participants of the European Prospective Investigation into Cancer and Nutrition (EPIC) study from 10 European countries	All incident EC cases were included based on self-report by questionnaire; follow-up data obtained via national cancer registries or insurance databases	HR 2.52; 95% CI 1.77–3.57 for ever users of estrogen-only MHT vs. never users	-
Razawi [7]	2010	CCS	311 (104)	570 (108)	Public school teachers and administrators; residence in California; data acquisition via California Cancer Registry	All histological types of EC included; exclusion of: in-situ carcinomas, endometrial sarcomas and mixed muellerian tumors	OR 4.46; 95% CI 2.46–8.11 for long-term (≥10 years) of estrogen-only MHT	OR 2.42; 95% CI 1.56–3.77 for <5 years OR 2.48; 95% CI 1.23–5.01 for 5–9 years
Doherty [26]	2007	CCS	1038 (341)	1453 (179)	US residents in 3 counties of Washington state; cases identified via registry (Cancer Surveillance System); controls recruited via random-digit-dialing telephone interview	All EC cases were included except for in situ and non-epithelial cases	OR 4.6; 95% CI 3.6–5.9 for ever users of estrogen-only MHT vs. never users	OR 11; 95% CI 7.7–15 for ever users of estrogen-only MHT for ≥6 years vs. never users
Lacey [27]	2007	CS	433	73,211	Members of American Association of Retired Persons (AARP) from 6 US states; data acquisition by repeated questionnaires	Incident EC cases identified from state-specific cancer registries and National Death Index	RR 1.23; 95% CI 0.89–1.82 for ever users of estrogen-only MHT vs. never users	RR 4.07; 95% CI 2.27–7.31 for long-term users (≥10 years) of estrogen-only MHT vs. never users
Strom [29]	2006	CCS	511 (35)	1412 (71)	US residents in Philadelphia region from 61 clinical centers; controls recruited via random-digit-dialing telephone interview	Only adenocarcinomas of the endometrium included; excluded: mixed muellerian tumors, sarcomas, undifferentiated carcinomas, and squamous cell carcinomas	OR 1.40; 95% CI 0.86–2.26 for ever use of estrogen-only MHT of any duration	OR 3.40; 95% CI 1.40–8.30 for estrogen-only MHT for >3 years
Beral [30]	2005	CS	1320	716,738	UK residents participating in the Million Women Study; data acquisition by repeated questionnaires	Incident EC cases were prospectively identified; median follow-up was 3.4 years	RR 1.45; 95% CI 1.02–2.06 for ever users of estrogen-only MHT vs. never users	Adverse effect of estrogen-only MHT was greatest among non-obese women
Bakken [32]	2004	CS	75	27,621	National, population-based cohort from Norway; postmenopausal women	Incident EC cases identified from questionnaires and followed-up using a national cancer registry; no restrictions on EC diagnosis	RR 3.2; 95% CI 1.2–8.0 for ever users of estrogen-only MHT vs. never users	-
Jain [35]	2000	CCS	512 (77)	513 (54)	Ontario, Canada	EC diagnosis restricted to adenocarcinoma, carcinoma, cystadenocarcinoma, or mixed Mullerian carcinoma; cases identified by Ontario Cancer Registry	OR 2.23; 95% CI 1.45–3.43 for ever use of estrogen-only MHT vs. never use	OR 4.12; 95% CI 2.21–7.71 for >3 years of estrogen-only MHT vs. never use
*Pooled Analysis*	-	CCS (*n* = 3), CS (*n* = 7)	10,674 (10,241 after exclusion of update studies)	2,029,655 (1,952,004 after exclusion of update studies)	-	-	Increased risk: 9 studies; no increased risk: 3 studies	Increased risk with long-term use (7 studies)

Abbreviations: MHT, menopausal hormone therapy; EC, endometrial cancer; CCS, case–control study; OR, odds ratio; CI, confidence interval; CS, cohort study; HR, hazard ratio; CEE, conjugated equine estrogens; US, United States; UK, United Kingdom; RR, relative risk; IRR, incidence rate ratio.

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
