# Peer review of "Menopausal Hormone Therapy and Risk of Endometrial Cancer: A Systematic Review"

_cancers, 2020, doi:10.3390/cancers12082195_

Round 1
Reviewer 1 Report
The authors responded satisfactorily to the comments and improved the quality of the manuscript.
Author Response
Response: no further action necessary.
Reviewer 2 Report
In my previous review, I suggested to the authors to improve the references of the whole manuscript because not all relevant works have been listed. In particular, a more accurate bibliography was necessary for the introduction section in order to improve and to update the scientific background.
Although the discussion section was improved with several references to support the assertions, the authors didn’t address my concerns.
So, in my opinion, it is a good manuscript, well written but not so updated for a review article.
However, in the present form, the manuscript can be accepted.
Author Response
No further action necessary. The reviewer states that: ,...in the present form, the manuscript can be accepted...'
Reviewer 3 Report
The authors mention “the latest review on this topic dates back to 2016”.
Among the reference list of the manuscript, there are ~6 papers that were published after 2016. Among them, only 3 of them were primary research papers, others were reviews, meaning that may be around 3 primary research papers that are newly reviewed by this current review. Therefore, it is believed that this review is reviewing a rather old topic that has been reviewed for many times. The novelty of this review is low and not much new information can be added. Not much new direction is revealed as well.
Author Response
Reviewer’s comment:
The authors mention “the latest review on this topic dates back to 2016”. Among the reference list of the manuscript, there are ~6 papers that were published after 2016. Among them, only 3 of them were primary research papers, others were reviews, meaning that may be around 3 primary research papers that are newly reviewed by this current review. Therefore, it is believed that this review is reviewing a rather old topic that has been reviewed for many times. The novelty of this review is low and not much new information can be added. Not much new direction is revealed as well.
Authors‘ Response:
The reviewer is not correct. Our literature search identified the following review articles/meta analyses (page 8, line 171):
,…2 narrative reviews [15, 31], 1 meta-analysis [22], and 1 systematic review [13]…’
These 4 citations have been published in 2014 [15], in 2005 [31], in 2010 [22], and in 2016 [13] as correctly cited in the text.
The criticism regarding the lack of novelty, however, is also invalid. The topic was not chosen by the authors bit by the Editorial Board of ,Cancers’. Hormone therapy clearly is a risk factor of endometrial cancer. Therefore, this topic must be included in a Special issue on ,Risk Factors of Endometrial Cancer’ irrespective of its degree of novelty.
This manuscript is a resubmission of an earlier submission. The following is a list of the peer review reports and author responses from that submission.
Round 1
Reviewer 1 Report
The review topic is a relatively old topic and there may be need to reveal what is the new information generated by the analysis in this review.Reviewer 2 Report
- It is a comprehensive work. However, it is not a systematic review. Although the authors did a "systematic" check of the literature, the manuscript cannot be addressed as a systematic review (SR) because it does not follow the criteria for SR.
- I suggest you follow the PRISMA guidelines if they want to do a systematic review. In this sense, it is necessary to explicitly include the research question following the PICO format. In addition to including a checklist table according to the PRISMA guide (as supplementary material).
- Or, instead, remove the "systematic review" and submitted as a review paper.
- Superiority and dissimilarity of this study from the other studies have not been emphasized enough. The difference should be emphasized.
- In the discussion, some paragraphs are not supported by references. For example the first paragraph.
- What are the limitations of the study? What are the recommendations for future studies?.
- In the conclusion, the main reliable results should be indicated more broadly and precisely, highlighting the studies that report the sizes of risk of clinical importance (OR> 3).
Reviewer 3 Report
MANUSCRIPT REVIEW
In the study entitled “Menopausal hormone therapy and risk of endometrial cancer: a systematic review” Tempfer et al. performed a systematic literature search of the databases PubMed and Cochrane Central Register of Controlled Trials to highlight controlled and uncontrolled clinical trials reporting on the prevalence and/or incidence of EC among women using MHT; they identified 31 publications reporting on 21,306 women with EC diagnosed during or after MHT. On the basis of their results, the authors concluded that ccMHT with SPs reduces the risk of EC, whereas estrogen-only MHT increases the risk. Also, scMHT with SPs and cc/scMHT with micronized progesterone increase the risk of EC depending on type of progestin, progestin dosage, and duration of MHT use.
The review is interesting, well written and the conceptional structure is well organized.
However, I recommend minor revision because the Introduction needs to be improved. Also, not all relevant works have been listed in the “References” section.
Point by point
INTRODUCTION
The conceptual structure of introduction is well organized but I suggest more accurate bibliography research because not all relevant works have been listed. In this way, you could develop this section for a better and most up to date background.
REFERENCES
You should add more references because not all relevant published works have been cited.
Reviewer 4 Report
This a systematic review on the impact of hormone therapy in the risk of endometrial cancer during menopause.
As the authors mention, there are other previous studies with similar conclusions; so originality is an issue, although the paper is an updated version.
Results show a reduced risk of endometrial cancers with continuous-combined therapy, in comparison with estrogen-only treatment. For sequential combined therapy or continuous-combined/sequential combined with micronized progesterone, the risk depends on type, dosage, and duration of progestins.
As it is, the paper has limited interest. As mentioned by the authors, there are different types of endometrial cancer, with different relation to estrogen stimulation, being endometrioid carcinoma the type of tumor more closely associated- Description of tumor characteristics is among the information available for a significant proportion of the articles. The paper would benefit from an in-deep description of the type of tumor associated with the different types of hormonal therapy.
Moreover, endometrial cancer occurs in patients under tamoxifen treatment of endometrial cancer. Tamoxifen has anti- and pro-estrogenic effects depending on the context. It would be nice to compare the results of hormonal therapy with those of tamoxifen treatment.